# Validation and reliability of the Bahasa Malaysia language version of the Acceptance of Illness Scale among Malaysian patients with cancer

Wun Chin Leong[1,2], Nor Aniza Azmi[1], Lei Hum Wee[1], Harenthri Devy Alagir Rajah[1], Caryn Mei Hsien Chan[1]*

1 Faculty of Health Sciences, Universiti Kebangsaan Malaysia, Kuala Lumpur, Malaysia, 2 Department of Radiotherapy and Oncology, National Cancer Institute, Putrajaya, Malaysia

* caryn@ukm.edu.my

## Abstract

Cancer is a life-threatening disease, and the challenges in accepting the diagnosis can bring a devastating emotional impact on the patient's mental and psychological wellbeing. Issues related to illness acceptance among cancer patients are not well studied in Malaysia. To date, the Acceptance of Illness Scale has not been translated to the Malay language *(Bahasa Malaysia)* nor validated for use in the oncology setting. The objective of the study is to translate, validate and determine the reliability of the Bahasa Malaysia version of the Acceptance of Illness Scale among Malaysian patients with cancer. A total of 129 patients newly diagnosed with cancer were consecutively sampled and the scale was administered via face-to-face interviews. A pilot test (n = 30) was conducted and test-retest reliability was determined. The Bartlett Test of Sphericity was statistically significantly (p<0.001), while the Kaiser-Mayer-Olkin (KMO) measure of sampling adequacy was adequate at 0.84. Scale item mean scores ranged between 3.02 and 4.33, while the item-total correlation ranged between 0.50 to 0.66 (p<0.05). The internal reliability coefficient was 0.84. The test-retest reliability indicated a high correlation, r = 0.94 with p = 0.001. The Bahasa Malaysia version of the Acceptance of Illness Scale is a valid and reliable instrument that is appropriate for use in Malaysian patients with cancer. Use of this scale to assess illness acceptance among the Malay-speaking patients with cancer can act as a guide for delivery of psycho-oncological services to help patients have a better mental wellbeing and life adjustment in living with cancer.

## Introduction

According to the World Health Organization, the global cancer burden will increase to 30 million by the year 2040, which is double from the estimated number of 18.1 million in 2018 [1]. In Asia, the number of the new cancer cases is projected at 10.6 million cases by 2030 [2]. Cancer and the accompanying side effects of treatment can intensify negative emotions which is

(Phone: 03-2615 6391) ; Address: Jalan Pahang, 50586 Kuala Lumpur, Wilayah Persekutuan Kuala Lumpur), University Malaya Medical Center (Email: ummc@ummc.edu.my (Phone : 03-7949 4422) ; Address : Jalan Profesor Diraja Ungku Aziz, 59100 Kuala Lumpur, Selangor) and Institut Kanser Negara (Email: ncipro@nci.gov.my (Phone : 03-8892 5555) ; Address : 4, Jalan P7, Presint 7, 62250 Putrajaya, Wilayah Persekutuan Putrajaya) respectively for permission to use and obtain patient data from this study. Ethical approval should also be sought separately for each of the three centers. Approval from the Director General of Health, Malaysia is also required. We confirm that the authors had no special access to privileges that others would not have.

**Funding:** This is funded by the Malaysia Research University Network (MRUN)-Long Term Research Grant (LRGS) and Universiti Kebangsaan Malaysia grants NN-2019-090 and DIP-2018-035. The funders had no role in the study design, collection of data and analysis, manuscript preparation or decision to publish. All the authors declared there is no following financial interest/personal relationship which may be considered as potential competing interests.

**Competing interests:** The authors have declared that no competing interests exist.

associated with the patient's acceptance of illness [3]. Acceptance of illness is important in people living with chronic diseases, including patients with cancer, as it influences a patient's attitude and subsequent coping strategy [4]. Illness acceptance can affect various aspects of a patient's life, such as their physical, mental, emotional, social and spiritual wellbeing [4–6], and may serve as a psychological proxy of disease adaptation. Patients with low levels of illness acceptance are more likely to have higher negative emotions and lower levels of adaptation, and therefore, also more likely to withdraw from recommended cancer treatments [7].

The Acceptance of Illness Scale is a tool designed to measure a patient's acceptance of his or her illness [8]. The scale comprises eight items that describe the difficulties and limitations associated with the negative effects of a poor health status. The limitations caused by an illness can lead patients to feel dependent on others, resulting in feelings of lack of self-sufficiency, low self-esteem, and poorer psychological sequelae in general [9–11]. The available evidence on illness acceptance among patients with cancer indicate that the poorer the illness acceptance, the more severe the restrictions in patient adjustment in living with their disease [2, 12]. Conversely, greater acceptance of illness has been found to be associated with less negative emotions and higher levels of motivation to seek treatment [13–15].

Given that the national lingua franca in Malaysia is the Malay language (*Bahasa Malaysia*), a translated instrument would be both important and useful to evaluate the level of illness acceptance among patients with cancer in Malaysia. We therefore aimed to translate and validate a Bahasa Malaysia version of the Acceptance of Illness scale, and to determine the psychometric properties of the translated scale.

## Method

### Study participants

Consecutive sampling was used. Participants were recruited from the Kuala Lumpur Hospital, National Cancer Institute and University Malaya Medical Centre. The inclusion criteria were as follows: 1) patients aged at least 18 years old and above, 2) able to understand and converse in Bahasa Malaysia, and 3) newly diagnosed with any cancer and first presentation at the oncology clinic. Patients who were too ill to tolerate an interview were excluded from the study.

### Ethical consideration

Permission to translate and validate the Acceptance of Illness Scale was sought and obtained from the developer of the scale, Tracey A. Ravenson. Ethical approval was obtained from the Malaysian Medical Research & Ethics Committee (MREC; NMRR-19-1118-45622 (IIR). All study participants provided written informed consent.

### Data collection procedures

The data was collected by trained graduate research assistants with backgrounds in psychology. Newly diagnosed patients with cancer were approached at the respective outpatient oncology clinics at each study site. Patients were screened to determine if they met all the inclusion and exclusion criteria. Patients who met the eligibility criteria were then informed about the study's objective, methods and the option to withdraw at any stage. Written informed consent was obtained. The study questionnaire was administered via face-to-face interviews.

### Instrument

The Acceptance of Illness Scale consists of eight items and is answered based on a five-point Likert scale. The participants evaluated each item on a scale from 1 (very poor acceptance to

illness), 2 (poor acceptance to illness), 3 (average acceptance to illness), 4 (acceptance to illness), and 5 (fully acceptance to illness). The level of the acceptance of illness is measured by summing up the scores from the individual statements, which ranges between 8 to 40. The higher the score obtained, the higher the level of acceptance of illness [12]. In this study, the cut-off point was based on the original tools designed by Felton et al. [20], participants were divided into three groups depending on the calculated Acceptance of Illness Scale score: (8–18 points) low acceptance of illness, (19–29 points) average acceptance of illness and (30–40 points) good acceptance of illness.

## Translation process

The English version of the Acceptance of Illness Scale was translated into Bahasa Malaysia based on the guidelines from the EORTC Quality of Life Group manual [16]. The translation was carried out by three independent experts, including a psychologist and a linguistic professional. All were proficient in both English and Bahasa Malaysia and had no prior knowledge of the scale. Back-to-back translation was also conducted. The translated Bahasa Malaysia version was then reviewed by two Bahasa Malaysia language native speakers, who compared the translated version to the original English version and made the necessary changes to the Bahasa Malaysia version [17].

Next, a pilot test was conducted with the cancer patients (n = 30). In the pilot, respondents were administered the translated Bahasa Malaysia version of Acceptance of Illness Scale and asked to provide feedback on the questions through a semi-structured interview which lasted approximately 10 to 15 minutes [16]. The purpose of this was to verify the comprehension of the scale by the respondents. As the translated items were deemed easily understandable, the translated Bahasa Malaysia version of the scale was then tested using psychometric analyses to determine its validity and reliability.

## Data analysis

Data were statistically analysed using SPSS version 26. Descriptive statistics were used to tabulate participants' socio-demographic characteristics using frequencies, means, and standard deviations. To determine the validity of the Bahasa Malaysia version of the Acceptance of Illness Scale, Principal Component Analysis (PCA) was used to examine the construct validity. The Bartlett's test of sphericity was used to ascertain sampling adequacy, internal consistency, factors rotation and factors identification. Regarding extraction, the Kaiser's criterion and the scree plot were assessed. To evaluate the test-retest reliability, Pearson correlation coefficients were calculated by comparing the scores at the test and retest phases. Cronbach's alpha was calculated to assess the internal consistency of the scale.

# Result

## Descriptive analysis

A total of 129 cancer patients were recruited for this validation study. Of these, 67.2% were female and 32.8% were male. The majority of the study population were of Malay ethnicity (81.3%), followed by the Chinese (8.6%) and the Indians (7.8%). Most of the participants were between 40–54 years old, with a mean age of 48 years (SD = 12.46). About 78.2% of the participants were married, 56.3% attained a secondary education, and 43.8% employed, with 65.1% hailing from low-income households. Most of the participants were diagnosed with breast cancer (33.3%), followed by gastrointestinal cancer (22.5%) and gynaecological cancer (14.0%). A total of 30.1% of the participants were diagnosed with stage II cancer, while 28.0% were

diagnosed with stage III cancer. The socio-demographic and clinical characteristics of the study participants are as shown in Table 1.

### Test-retest reliability analysis

The test-retest reliability for the Bahasa Malaysia version of the Acceptance of Illness Scale was assessed among 30 cancer patients. The translated scale was readministered two weeks after initial recruitment. The test-retest reliability using the Pearson Correlation Coefficient shows a high correlation, r = 0.94 with p = 0.001.

### Factors analysis

The Bartlett Test of Sphericity was statistically significant ($\chi^2$ = 366.321, p <0.001), while the Kaiser-Mayer-Olkin (KMO) measure of sampling adequacy was 0.84. Thus, factors analysis was deemed appropriate for this study. The principal component analysis (PCA) was used as the extraction method, and factors with Eigenvalues >1 according to Kaiser's criterion and Cattell's scree plot were retained for Varimax rotation with Kaiser normalization [18, 19]. Based on the PCA, two factors with Eigenvalues >1 were generated, which together explained 62.32% of the variance. The first component had an Eigenvalue of 3.87 and explained 48.42% of the variance, while the second component had an Eigenvalue of 1.112 and explained 13.91% of the variance are as shown in Table 2.

### Scree plot

The scree plot as shown in Fig 1 was used to examine the number of components above the inflection point.

The distribution of the descriptive item statistical values of the translated Acceptance of Illness Scale are as shown in Table 3. The mean score ranged between 3.02 and 4.33. The average mean score was 3.63, while the total mean score was 29.01 (SD 6.74).

### Internal consistency

Table 4 presents results from the internal consistency analysis. The lowest correlation was 0.50, while the highest was 0.66 (p<0.05). The item-total scale score correlation coefficient was 0.50, with no underlying negative related items.

To determine the internal consistency of the translated Acceptance of Illness Scale, the Cronbach's alpha coefficient was examined. The internal reliability coefficient for the total Cronbach's alpha was 0.84, and ranged between 0.81 and 0.83 for each item, indicating that the translated instrument shows good internal consistency. Table 5 shows the Cronbach's alpha values if the items of the translated Acceptance of Illness Scale were deleted. It can be observed that the alpha value did not undergo major changes if any of the items were removed from the scale.

## Discussion

This study sought to determine the validity and reliability of the translated Bahasa Malaysia version of the Acceptance of Illness Scale for use among patients with cancer in the Malaysian setting. The translated instrument was found to have good reliability, as well as content and construct validity. The internal consistency was found to be high (Cronbach's alpha = 0.81 to 0.83) and reflected good psychometric properties of the translated scale for use among Malaysian patients with cancer.

**Table 1. Patient socio-demographic characteristics (n = 129).**

| Characteristics | | N = 129 (%) |
|---|---|---|
| Age (Mean ± SD) | | 48.08±12.46 |
| 18–39 | | 33 (25.6) |
| 40–54 | | 56 (43.4) |
| 55–69 | | 34 (26.4) |
| 70 and above | 6 (4.7) | |
| Gender | | |
| Male | | 43 (32.8) |
| Female | | 86 (67.2) |
| Ethnicity | | |
| Malay | | 104 (81.3) |
| Chinese | | 11 (8.6) |
| Indian | | 10 (7.8) |
| Others | | 4 (2.3) |
| Marital status | | |
| Single | | 14 (11.0) |
| Married | | 100 (78.2) |
| Divorced | | 6 (4.7) |
| Widowed | | 8 (6.3) |
| Unknown | | 1 (0) |
| Education | | |
| Primary | | 16 (12.5) |
| Secondary | | 72 (56.3) |
| College/ University (Degree) | | 18 (14.1) |
| STPM/ Matriculation/ A level/ Diploma | | 21 (16.4) |
| Unknown | | 2 (0) |
| Employment status | | |
| Employed | | 56 (43.8) |
| Unemployed | | 43 (33.6) |
| Retired | | 20 (15.6) |
| Other | 9 (7.0) | |
| Unknown | | 1 (0) |
| Monthly household income | | |
| Less than RM 4,359 | | 82 (65.1) |
| RM 4,360—RM 9,619 | | 24 (19.0) |
| RM 9,620 and above | | 4 (3.2) |
| Others | | 16 (12.7) |
| Unknown | | 3 (0) |
| Setting | | |
| Kuala Lumpur Hospital | | 61 (48.8%) |
| National Cancer Institute | | 63 (47.3%) |
| University Malaya Medical Centre | | 5 (3.9%) |
| Cancer type | | |
| Breast | | 43 (33.3) |
| Gastrointestinal | | 29 (22.5) |
| Gynaecological | | 18 (14.0) |
| Lung | | 8 (6.2) |
| Sarcoma | | 5 (3.9) |

*(Continued)*

**Table 1.** (Continued)

| Characteristics | N = 129 (%) |
|---|---|
| Nose | 2 (1.6) |
| Haematological | 2 (1.6) |
| Brain | 1 (0.8) |
| Prostate | 2 (1.6) |
| Unknown | 19(14.7) |
| Cancer stage | |
| I | 14 (15.1) |
| II | 28 (30.1) |
| III | 26 (28.0) |
| IV | 25 (26.9) |
| Unknown | 36 (0) |

Previously, Felton et al. [20] used the acceptance of illness questionnaire (which included assessment of mood and social function) to measure the adjustment to illness in patients with chronic illness (namely hypertension, diabetes, arthritis, and cancer). The Acceptance of Illness Scale was originally developed in English. The tool has not been translated to Bahasa Malaysia and had not been validated among patients with cancer in Malaysia. To the best of our knowledge, this study is the first to utilise the Bahasa Malaysia version of the tool in a sample of patients with cancer in Malaysia and can be used as a future reference to accurately predict the patient's sense of adjustment to their illness at every stage of their disease.

According to the psychological stress and coping theories by Lazarus and Folkman [21], coping strategies enable patients to deal with distressing situations (problem-focused coping) and reduce negative emotions by avoiding harmful thoughts without changing the stressful situations (emotion-focused coping). Benson et al. [22] reported that the level of illness acceptance impacts patient's coping skills towards cancer. The higher the level of illness acceptance, the better the adaptation towards one's own health condition in response to emotional distress [7] and subsequently, patients developed better coping skills [2].

The factor analysis for the Acceptance of Illness Scale revealed that the Bartlett Test of Sphericity was statistically significant and the Kaiser-Mayer-Olkin measure of sampling adequacy was 0.84. According to Kaiser [18], KMO values that are greater than 0.8 are considered good. The mean score of the Acceptance of Illness in this sample was 29.01 (SD 6.74), demonstrating that the cancer patients in this study had a moderate level of illness acceptance. This is similar to the findings by Budna et al. [23] who reported a mean score of 28.30 among patients with

**Table 2. Total variance explained in the component PCA.**

| Component | Initial Eigenvalues | | | Extraction Sums of Squared Loadings | | | Rotation Sums of Squared Loadings | | |
|---|---|---|---|---|---|---|---|---|---|
| | Total | Variance % | Cumulative % | Total | Variance % | Cumulative % | Total | Variance % | Cumulative % |
| 1 | 3.873 | 48.415 | 48.415 | 3.873 | 48.415 | 48.415 | 3.160 | 39.505 | 39.505 |
| 2 | 1.112 | 13.906 | 62.321 | 1.112 | 13.906 | 62.321 | 1.825 | 22.816 | 62.321 |
| 3 | 0.780 | 9.745 | 72.066 | | | | | | |
| 4 | 0.608 | 7.603 | 79.669 | | | | | | |
| 5 | 0.501 | 6.261 | 85.930 | | | | | | |
| 6 | 0.442 | 5.527 | 91.457 | | | | | | |
| 7 | 0.356 | 4.456 | 95.913 | | | | | | |
| 8 | 0.327 | 4.087 | 100.000 | | | | | | |

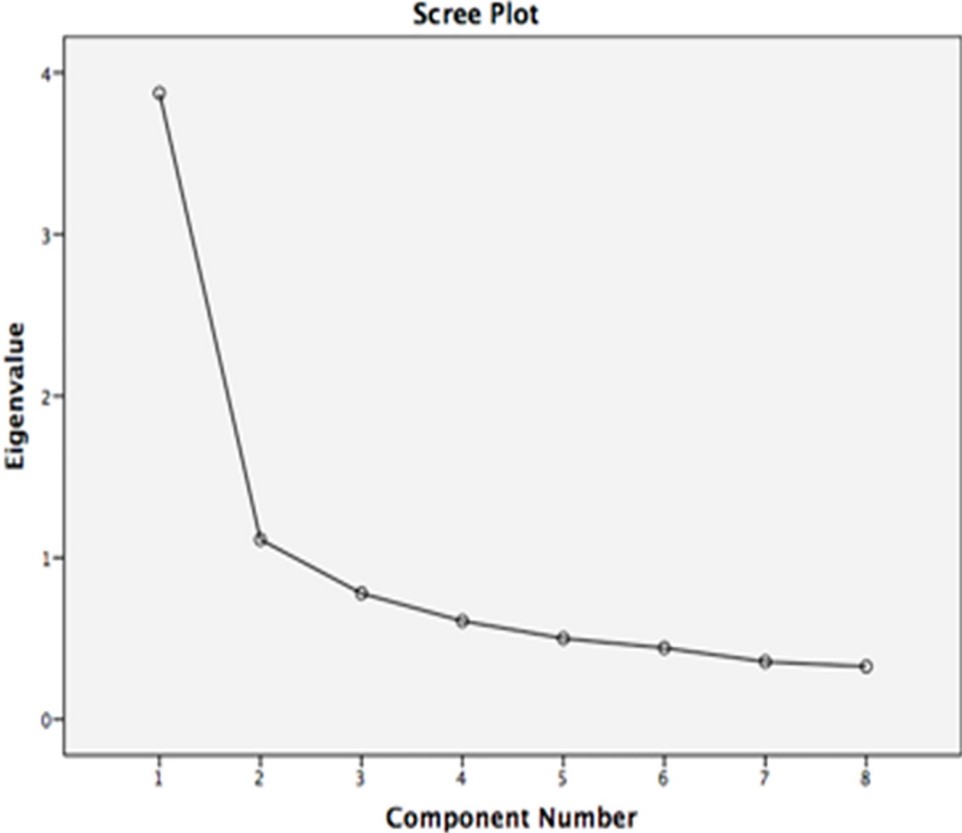

**Fig 1. Scree plot and point of inflection.**

cancer post-surgery. Conversely, Religioni et al. [24] reported a high level of illness acceptance with a mean score of 30.39 among patients diagnosed with lung, breast, colorectal and prostate cancer.

In terms of item analysis, if the correlation of an item has a low total score, it can be interpreted that the item measures a feature different from the other items. A low total correlation of the item might affect reliability, therefore those items should be removed from the scale. In this study, the item-total correlation coefficient on the Bahasa Malaysia version of the Acceptance of Illness Scale reliability was 0.50. The lowest score for the item-total correlation

**Table 3. Mean and standard deviations by item (n = 129).**

| Items | Mean | Standard Deviation (SD) |
|---|---|---|
| 1. I have problems with adjustment to the limitations imposed by the illness | 3.18 | 1.30 |
| 2. Due to my state of health I am not able to do what I like best | 3.02 | 1.35 |
| 3. The disease sometimes makes me feel unnecessary | 3.72 | 1.23 |
| 4. Because of health problems I am more dependent on others than I wish to be | 3.40 | 1.35 |
| 5. Due to the disease I am a burden on my family and friends | 3.58 | 1.30 |
| 6. Due to my health status I do not feel a fully valued human being | 4.09 | 1.07 |
| 7. I will never be self-sufficient to the degree I would like to be | 3.70 | 1.18 |
| 8. I think that people who stay with me are often embarrassed because of my illness | 4.33 | 0.96 |
| **Total Score** | 29.01 | 6.74 |

**Table 4. Examination of item-total score correlations of the Acceptance of Illness Scale (N = 129).**

| Items | Corrected Item-Total Correlation |
|---|---|
| 1. I have problems with adjustment to the limitations imposed by the illness | 0.50 (p< 0.05) |
| 2. Due to my state of health I am not able to do what I like best | 0.50 (p< 0.05) |
| 3. The disease sometimes makes me feel unnecessary | 0.57 (p< 0.05) |
| 4. Because of health problems I am more dependent on others than I wish to be | 0.59 (p< 0.05) |
| 5. Due to the disease I am a burden on my family and friends | 0.61 (p< 0.05) |
| 6. Due to my health status I do not feel a fully valued human being | 0.63 (p< 0.05) |
| 7. I will never be self-sufficient to the degree I would like to be | 0.66 (p< 0.05) |
| 8. I think that people who stay with me are often embarrassed because of my illness | 0.59 (p< 0.05) |

coefficient was 0.50 for item 1 ("I have problems with adjustment to the limitations imposed by the illness") and item 2 ("Due to my state of health I am not able to do what I like the most"), whereas the highest item-total score was 0.66 for item 7 ("I will never be as self-sufficient to the degree I would like to be"). The overall item-total scale score indicated that it was not necessary to remove any items as there seemed to be no underlying negative-related items.

Internal consistency of the Bahasa Malaysia version of the scale was found to be high in the current study. This is similar to that of the English version [20, 25–27] among patients with chronic illness (namely hypertension, diabetes, arthritis, and cancer). In this study, the Cronbach alpha value did not change excessively when the items were deleted. Thus, the current study showed that the translated scale had sufficient homogeneity with good reliability.

In the present study, the high test-retest reliability using the Pearson Correlation Coefficient indicate that the Bahasa Malaysia version of the Acceptance of Illness Scale can be beneficial for use in longitudinal studies, and can provide a good evaluation of illness acceptance at different time intervals. Our findings are in line with the original study that reported an acceptable test-retest reliability after seven months [20, 25], as well as with a study by Juczynski [26] which reported satisfactory test-retest reliability after a four week period. We conducted the test-retest reliability after a two week period, which is a shorter period compared to other studies, because in Malaysia oncology centres practice the two weeks' wait rule with newly diagnosed patients with cancer for their oncologist to review their treatment discussion and decision follow-up. This is to ensure patients have better self-emotion adjustment and mental preparedness before the start of treatment.

**Table 5. Cronbach's alpha values by item deletion for the translated Acceptance of Illness Scale.**

| Items | Cronbach's Alpha if Item Deleted |
|---|---|
| 1. I have problems with adjustment to the limitations imposed by the illness | 0.83 |
| 2. Due to my state of health I am not able to do what I like best | 0.83 |
| 3.The disease sometimes makes me feel unnecessary | 0.82 |
| 4. Because of health problems I am more dependent on others than I wish to be | 0.82 |
| 5. Due to the disease I am a burden on my family and friends | 0.82 |
| 6. Due to my health status I do not feel a fully valued human being | 0.82 |
| 7. I will never be self-sufficient to the degree I would like to be | 0.81 |
| 8. I think that people who stay with me are often embarrassed because of my illness | 0.82 |
| **Total Cronbach's Alpha** | 0.84 |

The strength of the present study lies in its good test-retest reliability, good internal consistency, and sampling adequacy. This study is however not without its limitations. While respondents were derived from the three major referral oncology centers in the country, this may not have been representative of all patients with cancer in Malaysia. This translated tool can however be used as an important patient reported instrument to measure the levels of illness acceptance among Malay-speaking patients with cancer in this setting, and thereby lead to a further push for patient-centered cancer care in Malaysia [28, 29].

## Conclusion

The translated Bahasa Malaysia version of the Acceptance of Illness Scale is a valid and reliable instrument that can be used to quantitatively assess illness acceptance, given its high test-retest reliability and validity. As Bahasa Malaysia is the most common written and spoken language in Malaysia, particularly as the population comprises of a Malay majority, we believe this translated and validated scale will prove useful in future research for the majority of patients with cancer in this setting.

## Supporting information

**S1 Appendix. Acceptance of Illness Scale (English version).**
(PDF)

**S2 Appendix. Skala penerimaan penyakit (Bahasa Malaysia version).**
(PDF)

## Acknowledgments

The authors thank all study participants, the Clinical Research Centre and the Radiotherapy and Oncology Departments of the Kuala Lumpur Hospital, National Cancer Institute, and University Malaya Medical Centre. The authors also extend their thanks to Professor Dr. Tracey A. Revenson, the developer of the scale for permission to translate and validate the Acceptance of Illness Scale.

## Author Contributions

**Conceptualization:** Wun Chin Leong, Nor Aniza Azmi, Lei Hum Wee, Caryn Mei Hsien Chan.

**Data curation:** Wun Chin Leong, Harenthri Devy Alagir Rajah.

**Formal analysis:** Wun Chin Leong, Nor Aniza Azmi, Harenthri Devy Alagir Rajah, Caryn Mei Hsien Chan.

**Investigation:** Wun Chin Leong, Harenthri Devy Alagir Rajah.

**Methodology:** Wun Chin Leong, Caryn Mei Hsien Chan.

**Project administration:** Wun Chin Leong, Harenthri Devy Alagir Rajah, Caryn Mei Hsien Chan.

**Resources:** Wun Chin Leong.

**Software:** Harenthri Devy Alagir Rajah.

**Supervision:** Nor Aniza Azmi, Lei Hum Wee, Caryn Mei Hsien Chan.

**Validation:** Wun Chin Leong.

Writing – **original draft:** Wun Chin Leong.

Writing – **review & editing:** Nor Aniza Azmi, Caryn Mei Hsien Chan.

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
