## [Decision Letter · Decision Letter 0]

8 Apr 2021

PONE-D-21-01297

Validation and Reliability of the Acceptance of Illness Scale among Malaysian patients with cancer: the Bahasa Malaysia version in Malaysia

PLOS ONE

Dear Dr. Chan,

Thank you for submitting your manuscript to PLOS ONE. After careful consideration, we feel that it has merit but does not fully meet PLOS ONE’s publication criteria as it currently stands. Therefore, we invite you to submit a revised version of the manuscript that addresses the points raised during the review process.

Please carefully address all comments raised, and also refer to the availability of the data according to PLOS ONE requirements. 

We look forward to receiving your revised manuscript.

Kind regards,

Stefan Hoefer

Academic Editor

PLOS ONE

Journal Requirements:

Reviewers' comments:

Reviewer's Responses to Questions

**Comments to the Author**

1. Is the manuscript technically sound, and do the data support the conclusions?

Reviewer #1: Partly

Reviewer #2: Yes

Reviewer #3: Yes

2. Has the statistical analysis been performed appropriately and rigorously? 

Reviewer #1: I Don't Know

Reviewer #2: Yes

Reviewer #3: Yes

3. Have the authors made all data underlying the findings in their manuscript fully available?

Reviewer #1: No

Reviewer #2: Yes

Reviewer #3: Yes

4. Is the manuscript presented in an intelligible fashion and written in standard English?

Reviewer #1: No

Reviewer #2: No

Reviewer #3: Yes

5. Review Comments to the Author

Reviewer #1: It is very important to have research measures that are appropriate for different contexts and available in the language of choice of research participants. This study sought to validate a BM version of the Acceptance of Illness scale with a sample of cancer patients.

I think this manuscript would benefit from work to make the aims and objectives clearer. From the abstract it reads that this is the validation of an existing BM, but the main text makes it clear that the student actually did the translations. It would also benefit from some detailed proofing as in the PDF I received there were many cases of words merged together.

In relation to the methods this study was carried out with recently diagnosed patients. I believe the original measure was developed using data from patients who were chronically ill. Could the authors explain why they chose the sample they did and how they took stage of illness into account?

Inclusion and exclusion criteria should be outlined.

In relation to the test retest, I was unsure if this was a separate study or a sub study. It would be helpful to clarify this.

Construct validity is mentioned in the discussion, but no data is presented on this – I would recommend this be added.

Reviewer #2: This paper aims to translate the Acceptance of Illness Scale to the local language (Bahasa Malaysia), and validate and assess the reliability of the translated scale for use among cancer patients in Malaysia. The research are methodologically sound, with appropriate methods and statistical tests. The conclusion derived is supported by the findings of the study.

However this manuscript needs major revision before it can be published in the journal.

1. The English language is not up to standard, it makes reading difficult and affect understanding of what the authors are trying to say. It is strongly recommended that this manuscript is sent for English language proofreading.

Methodology section:

2. Sample size of 129- how did the authors arrive at this number? Was there any sample size calculations done?

3. Need to explain who is Prof Tracey. Is she the owner/ developer of the scale?

4. Need to clearly specify the inclusion and exclusion criteria

5. Under instrument - Specify all the 5-degree scales ie 1 - poor adaptation to illness, 2 - ?, 3 - ?, 4 - ?, 5 - ?. Remove the 'for example 1 - definitely agree, 2 -agree........'

6. Under Instrument- shouldn't the overall score of acceptance range between 5-40? Why did the author states it as 8-40?

7. Under instrument - explain how cut-off points for high, average and low acceptance was obtained?

8. Under translation - specify translation of scale to Bahasa Malaysia is by who and how many people

9. Under translation - remove the word 'about' 129 patients . You need to be very objective on the number of respondents included in the study

Data analysis section:

10. Need to expand to include all the test that has been conducted ie pearson correlation coefficient etc

11. In the text mentioned chi square test, but the results of the test is not shown in results or discussed in discussion section

Results section:

12. No need to mention about prospective study (n=150)

13. Is there a reason why the test-retest analysis is conducted on only 30 patients and not all 129?

Discussion section:

14. KMO index is not part of the factor analysis test. It is done prior to conducting factor analysis to see the adequacy of dataset to conduct factor analysis. Please rephrase your sentence on KMO under discussion as well as abstract

15. Findings from Table 2 is not discussed at all under discussion section

16. There is a lot of redundancies/ repeated statements across the manuscript:

a) mentioned many many times that the scale consists of 8 items

b) mentioned few times that the scale can be used to measure level of acceptance of illness among cancer patients

under methodology

c) mentioned few times that scale is translated to Bahasa Malaysia under methodology

17. Repeated discussion on high internal consistencies with the same references quoted

18. Please check the formatting of tables

Reviewer #3: 1) This is an important manuscript for South East Asians as Bahasa Malaysia is the main language used in Malaysia and in addition, it may also be used in Indonesia as well which has a large population that uses Bahasa Melayu.

2) In the last paragraph of the Introduction, it would be useful for the author to describe the multi-ethnic and multi-lingual society of Malaysia, including the estimated users of Bahasa Malaysia as the national language and to highlight the necessity to have this scale translated and validated for the local population where English is not the main language used.

3) In terms of the descriptive analysis, the cohort in this study has 56.3% with secondary school education, 14.1% with university and 16.4 with college / diploma education, which seems to be quite a highly educated cohort. Is this comparable to the Malaysia population generally? Please comment either way.

4) In the discussion on page 14, this study had a 2 week test-retest reliability score which is a shorter period compared to other studies. Can the author comment on this as oncology newly diagnosed patients are usually seen within a much longer period to complete their treatment.

6. PLOS authors have the option to publish the peer review history of their article (what does this mean?). If published, this will include your full peer review and any attached files.

Reviewer #1: No

Reviewer #2: No

Reviewer #3: No

---

## [Author Response · Author response to Decision Letter 0]

17 Jun 2021

Reviewer 1: I have incorporated all of your suggestions into my revision. They were very helpful. Thank you.

Reviewer 2: I have incorporated all of your suggestions into my revision. Thank you for your help.

Reviewer 3: I have incorporated all of your suggestions into my revision. Thank you so much and they were very helpful.

---

## [Decision Letter · Decision Letter 1]

11 Jul 2021

PONE-D-21-01297R1

Validation and Reliability of the Acceptance of Illness Scale among Malaysian patients with cancer: the Bahasa Malaysia version in Malaysia

PLOS ONE

Dear Dr. Chan,

Thank you for submitting your manuscript to PLOS ONE. After careful consideration, we feel that it has merit but does not fully meet PLOS ONE’s publication criteria as it currently stands. Therefore, we invite you to submit a revised version of the manuscript that addresses the points raised during the review process.

We look forward to receiving your revised manuscript.

Kind regards,

Stefan Hoefer

Academic Editor

PLOS ONE

Journal Requirements:

Reviewers' comments:

Reviewer's Responses to Questions

**Comments to the Author**

1. If the authors have adequately addressed your comments raised in a previous round of review and you feel that this manuscript is now acceptable for publication, you may indicate that here to bypass the “Comments to the Author” section, enter your conflict of interest statement in the “Confidential to Editor” section, and submit your "Accept" recommendation.

Reviewer #2: All comments have been addressed

Reviewer #3: (No Response)

2. Is the manuscript technically sound, and do the data support the conclusions?

Reviewer #2: (No Response)

Reviewer #3: (No Response)

3. Has the statistical analysis been performed appropriately and rigorously? 

Reviewer #2: (No Response)

Reviewer #3: (No Response)

4. Have the authors made all data underlying the findings in their manuscript fully available?

Reviewer #2: (No Response)

Reviewer #3: (No Response)

5. Is the manuscript presented in an intelligible fashion and written in standard English?

Reviewer #2: (No Response)

Reviewer #3: (No Response)

6. Review Comments to the Author

Reviewer #2: (No Response)

Reviewer #3: Thank you for your revision. For comment no 4, your response was "In Malaysia oncology centres practices, newly diagnosed patients appointment will be given in 2 weeks time to review specialist for treatment decision and treatment plan. Patients then will be retest on their second appointment (2 weeks), while they were on the waiting to

see specialist for treatment decision and treatment plan follow up. This is to ensure that the patients have feelings of preparedness in dealing with cancer and coping prior treatment commence."

Please summarise this paragraph and add on to the discussion of the paper to highlight the reason why the re-test is 2 weeks and not longer.

7. PLOS authors have the option to publish the peer review history of their article (what does this mean?). If published, this will include your full peer review and any attached files.

Reviewer #2: No

Reviewer #3: No

---

## [Author Response · Author response to Decision Letter 1]

13 Jul 2021

Reviewer 3: I have incorporated all of your suggestion into my revision. Thank you for your help.

---

## [Decision Letter · Decision Letter 2]

3 Aug 2021

Validation and Reliability of the Acceptance of Illness Scale among Malaysian patients with cancer: the Bahasa Malaysia version in Malaysia

PONE-D-21-01297R2

Dear Dr. Chan,

We’re pleased to inform you that your manuscript has been judged scientifically suitable for publication and will be formally accepted for publication once it meets all outstanding technical requirements.

Kind regards,

Stefan Hoefer

Academic Editor

PLOS ONE

Additional Editor Comments (optional):

Reviewers' comments:

Reviewer's Responses to Questions

**Comments to the Author**

1. If the authors have adequately addressed your comments raised in a previous round of review and you feel that this manuscript is now acceptable for publication, you may indicate that here to bypass the “Comments to the Author” section, enter your conflict of interest statement in the “Confidential to Editor” section, and submit your "Accept" recommendation.

Reviewer #2: All comments have been addressed

Reviewer #3: (No Response)

2. Is the manuscript technically sound, and do the data support the conclusions?

Reviewer #2: (No Response)

Reviewer #3: (No Response)

3. Has the statistical analysis been performed appropriately and rigorously? 

Reviewer #2: (No Response)

Reviewer #3: (No Response)

4. Have the authors made all data underlying the findings in their manuscript fully available?

Reviewer #2: (No Response)

Reviewer #3: (No Response)

5. Is the manuscript presented in an intelligible fashion and written in standard English?

Reviewer #2: (No Response)

Reviewer #3: (No Response)

6. Review Comments to the Author

Reviewer #2: (No Response)

Reviewer #3: (No Response)

7. PLOS authors have the option to publish the peer review history of their article (what does this mean?). If published, this will include your full peer review and any attached files.

Reviewer #2: No

Reviewer #3: No

---

## [Editor Report · Acceptance letter]

20 Sep 2021

PONE-D-21-01297R2 

Validation and reliability of the Bahasa Malaysia language version of the Acceptance of Illness Scale among Malaysian patients with cancer 

Dear Dr. Chan:

I'm pleased to inform you that your manuscript has been deemed suitable for publication in PLOS ONE. Congratulations! Your manuscript is now with our production department. 

Kind regards, 

on behalf of

Dr. Stefan Hoefer 

Academic Editor

PLOS ONE